# GCIB: Graph Contrastive Information Bottleneck for Multi-Behavior Recommendation

**Likang Wu** [1]  **Zihao Chen** [2]  **Jianxin Zhang** [2]  **Sangqi Zhu** [2]
**Yuanyuan Ge** [2]  **Haipeng Yang** [2]  **Lei Zhang** [2]

## Abstract

With the rapid emergence of multi-behavior learning in recommender systems, leveraging auxiliary user behaviors has proven effective for mitigating target-behavior data sparsity. Yet auxiliary behavior graphs frequently contain noisy or irrelevant interactions that do not align with the target task, impeding the learning of accurate user and item embeddings. Moreover, the scarcity of direct supervision from the target behavior complicates the extraction of informative collaborative signals. In this paper, we introduce *GCIB (Graph Contrastive Information Bottleneck)*, a novel framework that denoises auxiliary behavior information and enriches target behavior representations at both the structural and feature levels. At the structural level, GCIB employs a **G**raph **I**nformation **B**ottleneck (GIB) objective to maximize mutual information between the denoised auxiliary graph and the target-behavior graph while minimizing mutual information with the original auxiliary graph. This formulation preserves task-relevant structural patterns and suppresses spurious interactions. At the feature level, we propose a cross-behavior **G**raph **C**ontrastive **L**earning (GCL) scheme in which denoised auxiliary features and target-behavior features serve as complementary views for both users and items. By contrasting these views, GCIB enriches sparse target-behavior representations with semantics distilled from auxiliary behaviors. Extensive experiments demonstrate that GCIB outperforms state-of-the-art baselines, highlighting its ability to learn noise-resilient and target-aware representations for multi-behavior recommendation.

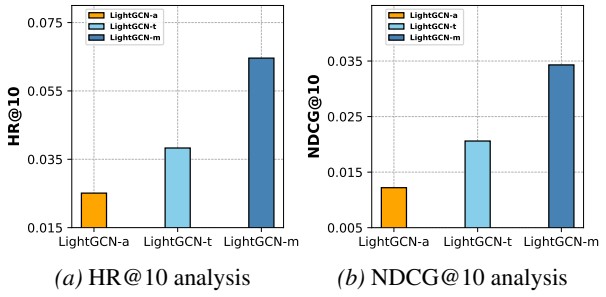

*(a) HR@10 analysis*        *(b) NDCG@10 analysis*

*Figure 1.* Performance comparison of LightGCN under different graph usage settings on the Tmall dataset

## 1. Introduction

In recent years, multi-behavior recommendation has emerged as a powerful solution to alleviate the data sparsity issue inherent in traditional single-behavior algorithms (Gu et al., 2022; Chen et al., 2020; Yan et al., 2023). Unlike single-behavior settings that rely solely on a single type of user interaction (e.g., clicks or purchases), multi-behavior frameworks incorporate a variety of user behaviors—such as views, likes, and add-to-cart actions—to enrich user preference modeling. These auxiliary signals provide valuable contextual information that helps uncover users' underlying intents, especially when target behavior data is insufficient.

Motivated by the success of graph neural networks (GNNs) in learning expressive representations from relational data, many recent studies have adopted GNN-based architectures for multi-behavior recommendation. These methods typically construct separate graphs for different behaviors, encode user-item interactions using GNN layers, and fuse the multi-behavior representations to enhance recommendation performance. Common strategies include behavior-specific graph modeling (Jin et al., 2020; Chen et al., 2021), attention-based behavior fusion (Wang et al., 2019b; Guo et al., 2019), or joint embedding learning across behaviors (Gao et al., 2019). While these approaches have achieved considerable improvements, they primarily focus on integrating all available behavioral data indiscriminately.

Despite their effectiveness, existing GNN-based multi-behavior recommendation models still suffer from limitations (Cheng et al., 2023; Xia et al., 2021). First, they

---
[1]Tianjin University, Tianjin, China [2]Anhui University, Hefei, China. Correspondence to: Lei Zhang <zl@ahu.edu.cn>.

*Proceedings of the $43^{rd}$ International Conference on Machine Learning*, Seoul, South Korea. PMLR 306, 2026. Copyright 2026 by the author(s).

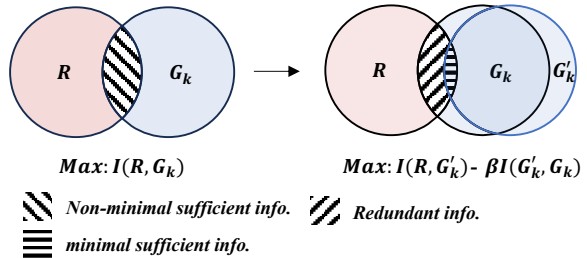

$$Max: I(R, G_k) \qquad Max: I(R, G_k') - \beta I(G_k', G_k)$$

▨ *Non-minimal sufficient info.*    ▨ *Redundant info.*

▤ *minimal sufficient info.*

*Figure 2.* Difference between traditional methods and GCIB in information utilization

often overlook the fact that auxiliary behavior graphs contain task-irrelevant or noisy information, which misleads the learning of target behavior preferences. Second, the inherent sparsity of target behavior interactions weakens the supervision signal needed for robust representation learning. To empirically demonstrate these issues, we conduct a comparative analysis on a benchmark dataset using three settings: (1) using only the target behavior graph (denoted as "-t"), (2) using only auxiliary behavior graphs (denoted as "-a"), and (3) using all behavior graphs jointly (denoted as "-m"). As shown in Figure 1, we observe that: (i) using only auxiliary behaviors yields the worst performance, confirming that auxiliary graphs include irrelevant or noisy structural patterns; (ii) using only the target graph leads to moderate performance, limited by sparse supervision; and (iii) combining all behaviors offers the best performance, indicating the benefit of multi-behavior integration, albeit with only marginal gains due to the presence of noise. These results suggest that **how to effectively denoise and leverage auxiliary behaviors remains a critical challenge** in multi-behavior recommendation.

Several recent works have attempted to address this challenge by denoising auxiliary data (Yang et al., 2025; Liu et al., 2023) or refining behavior-specific representations (Meng et al., 2023b). However, most existing methods either operate on a single modeling level or apply denoising after behavioral signals have already been propagated into latent representations. In particular, prior information bottleneck based recommendation methods mainly focus on representation-level compression, such as reducing redundancy in social representations, multimodal features, or hierarchical behavior embeddings. They do not explicitly optimize which auxiliary behavior edges should be retained before graph propagation. As a result, noisy auxiliary interactions may still participate in message passing and contaminate the learned user/item embeddings.

To address these limitations, we propose a novel framework called GCIB (Graph Contrastive Information Bottleneck) for multi-behavior recommendation. Our method performs *denoising from two complementary perspectives*: graph structure level and feature representation level. At the structure level, GCIB introduces an information bottleneck formulation to retain the minimal yet sufficient

substructure of the auxiliary behavior graph that is most relevant to the target recommendation task. Specifically, we optimize a mutual information objective of the form Maximize: $I(\mathcal{R}; \mathcal{G}_k') - \beta I(\mathcal{G}_k'; \mathcal{G}_k)$, where $\mathcal{G}_k$ is the original auxiliary behavior graph, $\mathcal{G}_k'$ is its denoised counterpart, and $\mathcal{R}$ represents the target behavior signal. This formulation, as illustrated in Figure 2, distinguishes our approach from prior methods that simply maximize $I(\mathcal{R}; \mathcal{G}_k)$ without suppressing redundant information. At the feature level, GCIB employs a *cross-behavior contrastive learning* mechanism that treats denoised auxiliary and target behavior features as semantically aligned views on both user and item sides. This enables GCIB to enrich sparse target representations while ensuring noise-resilient feature learning.

Implementing such a dual-denoising framework presents several challenges. First, it is non-trivial to formalize and optimize mutual information terms in a fully differentiable and unsupervised manner. Second, the absence of ground-truth labels for relevant auxiliary behaviors complicates the supervision of the denoising process. Third, designing effective cross-behavior contrastive strategies requires careful coordination of structural and semantic alignment across behaviors with inherently different interaction semantics.

To overcome these challenges, we develop a dual-perspective denoising framework with technical innovations at both the structure and feature levels. At the structure level, we first derive a variational lower bound of $I(\mathcal{R}; \mathcal{G}_k')$ to enable tractable optimization and guide the model to preserve target-relevant information in the refined graph. Meanwhile, instead of directly minimizing $I(\mathcal{G}_k'; \mathcal{G}_k)$, which is difficult to estimate in practice, we introduce the Hilbert-Schmidt Independence Criterion (HSIC) as a surrogate objective. This facilitates independence regularization between the original and refined auxiliary graphs, thereby suppressing task-irrelevant noise. Furthermore, to better guide the denoising process, we leverage users' preferences inferred from the target behavior data as supervision signals. At the feature level, we formulate a cross-behavior contrastive learning strategy, which semantically aligns the denoised auxiliary behavior representations with the target behavior features. This effectively supplements the sparse target behavior supervision with complementary auxiliary semantics while remaining resilient to noise. By jointly optimizing structural bottleneck and feature alignment objectives, our framework effectively removes irrelevant structures in auxiliary behavior graphs and enhances the representation quality under sparse target supervision. In summary, our main contributions are as follows:

- We propose a novel multi-behavior recommendation framework, GCIB, that performs dual-perspective denoising by jointly optimizing graph structure refinement and feature-level semantic alignment.

- We design a principled graph information bottleneck objective, where a variational lower bound of $I(\mathcal{R}; \mathcal{G}'_k)$ is maximized and HSIC is used to regularize redundancy between $\mathcal{G}'_k$ and $\mathcal{G}_k$.

- We develop a cross-behavior contrastive learning strategy that aligns denoised auxiliary features with target behavior representations, supplementing sparse supervision and improving recommendation accuracy.

The related work section can be found in Appendix A.

## 2. Preliminaries

### 2.1. Problem Definition

In multi-behavior recommendation, let $\mathcal{U} = \{u_1, \ldots, u_M\}$ denote the user set and $\mathcal{I} = \{i_1, \ldots, i_N\}$ denote the item set. Multiple user-item interactions are represented by a collection of behavior-specific interaction matrices $\{\mathcal{R}^{(k)} \in \mathbb{R}^{M \times N}\}_{k=1}^{\mathcal{K}}$, where $\mathcal{R}_{u,i}^{(k)} = 1$ indicates that user $u$ has performed behavior type $k$ (e.g., click, cart, purchase) on item $i$, and $\mathcal{R}_{u,i}^{(k)} = 0$ otherwise.

To jointly model heterogeneous interactions, the data can be represented as a heterogeneous bipartite graph $\mathcal{G} = \{\mathcal{V} = \mathcal{U} \cup \mathcal{I}, \mathcal{E}^{(1)}, \ldots, \mathcal{E}^{(\mathcal{K})}\}$, where all users $\mathcal{U}$ and items $\mathcal{I}$ are viewed as node set $\mathcal{V}$, and each edge set $\mathcal{E}^{(k)}$ corresponds to a specific behavior type. The objective is to learn behavior-aware user and item representations that capture cross-behavior dependencies and accurately predict the target behavior.

For each behavior graph $\mathcal{G}^{(k)}$, the $l$-th layer node embeddings are computed via a behavior-specific aggregation:

$$\mathbf{E}^{(k),l} = f_{\text{agg}}^{(k)}(\mathbf{E}^{(k),l-1}, \mathcal{G}^{(k)}), \tag{1}$$

where $\mathbf{E}^{(k),0}$ denotes the shared initial embeddings. After $L$ layers of propagation, embeddings from all behaviors are fused through a readout function:

$$\mathbf{E} = f_{\text{readout}}(\{\mathbf{E}^{(k),l}\}_{k=1,l=0}^{\mathcal{K},L}), \tag{2}$$

which integrates multi-behavior information via weighted or attention-based fusion to produce unified representations.

### 2.2. Information Bottleneck and Hilbert-Schmidt Independence Criterion

The Information Bottleneck (IB) principle aims to learn representations that retain task-relevant information while discarding irrelevant input signals. Formally, for input data $X$, hidden representation $Z$, and target label $Y$ following the Markov chain $X \to Z \to Y$, the IB objective is:

$$Z^* = \arg \max_Z I(Y; Z) - \beta I(X; Z), \tag{3}$$

where $I(Y; Z)$ measures the relevance of $Z$ to the task, $I(X; Z)$ quantifies the retained input information, and $\beta$ balances compression and sufficiency. IB has been applied in areas such as robustness (Wang et al., 2021; Wu et al., 2020), fairness (Gronowski et al., 2023), and explainability (Bang et al., 2021). In this work, we leverage IB to guide robust social denoising for recommendation.

To practically estimate mutual information in the IB framework, we adopt the Hilbert-Schmidt Independence Criterion (HSIC (Gretton et al., 2005)) as a surrogate metric. HSIC is a kernel-based statistical dependence measure, defined using the Hilbert-Schmidt norm of the cross-covariance operator between distributions embedded in a Reproducing Kernel Hilbert Space (RKHS). For two variables $X$ and $Y$, the HSIC is computed as:

$$
\begin{aligned}
HSIC(X, Y) &= \|C_{XY}\|_{hs}^2 \\
&= \mathbb{E}_{X,X',Y,Y'}[K_X(X, X')K_Y(Y, Y')] \\
&\quad + \mathbb{E}_{X,X'}[K_X(X, X')]\mathbb{E}_{Y,Y'}[K_Y(Y, Y')] \\
&\quad - 2\mathbb{E}_{XY}[\mathbb{E}_{X'}[K_X(X, X')]\mathbb{E}_{Y'}[K_Y(Y, Y')]],
\end{aligned}
\tag{4}
$$

where $K_X$ and $K_Y$ are the kernel functions for $X$ and $Y$, and $(X', Y')$ denotes an independent copy of $(X, Y)$.

Given a batch of training samples $\{(x_i, y_i)\}_{i=1}^n$, the empirical estimation of HSIC is given by:

$$\hat{HSIC}(X, Y) = (n-1)^{-2}\text{Tr}(K_X H K_Y H), \tag{5}$$

where $K_X$ and $K_Y$ are the kernel Gram matrices, $H = \mathbf{I} - \frac{1}{n}\mathbf{1}\mathbf{1}^\top$ is the centering matrix, and $\text{Tr}(\cdot)$ denotes the trace operator. In practice, we employ the Radial Basis Function (RBF (Vert et al., 2004)) kernel:

$$K(x_i, x_j) = \exp\left(-\frac{\|x_i - x_j\|^2}{2\sigma^2}\right), \tag{6}$$

where $\sigma$ is the bandwidth parameter that controls the sharpness of the kernel.

By leveraging HSIC within the IB framework, we approximate and optimize the mutual information terms in a differentiable manner, enabling efficient learning of task-relevant and noise-resistant representations for recommendation.

## 3. Methodology

### 3.1. Overview of GCIB

The overall framework of GCIB is illustrated in Figure 3. The model adopts an end-to-end collaborative training paradigm. A heterogeneous multi-behavior graph (MBG) is first constructed from user-item interactions, where domain-aware modeling captures distinct behavioral patterns (e.g., click, cart, purchase) and heterogeneous adjacency matrices encode unified cross-behavior dependencies. A hierarchical behavior decoupling module with an information bottleneck-guided pruning mechanism then filters out noisy edges and

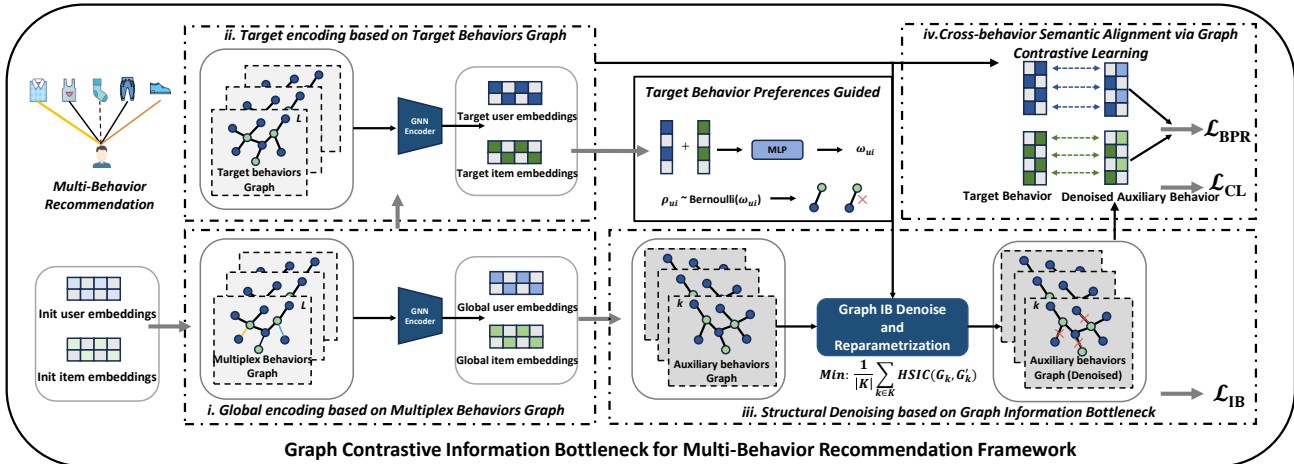

*Figure 3.* The overall structure of the presented GCIB model

enables GCNs to learn target-aware representations on the refined subgraph. To ensure semantic consistency across behaviors, GCIB introduces a contrastive alignment module that applies semantic-preserving augmentation to the target graph and leverages denoised auxiliary graphs as contrastive views for adaptive representation alignment. Finally, decoupled auxiliary behavior embeddings and aligned target semantics are fused for prediction. This integrated framework mitigates noise through dynamic pruning, enhances discriminability via semantic-aware contrastive learning, and promotes structural-semantic knowledge transfer for improved recommendation accuracy.

## 3.2. Global Encoding Based on Heterogeneous Multi-Behavior Graphs

As shown in the first part of Figure 3, to better leverage the relationship between auxiliary behaviors and target behaviors, we propose global encoding based on heterogeneous multi-behavior graphs, serving as the model's initial embedding representation. First, a global heterogeneous multi-behavior graph $\mathcal{G}_{global} = (\mathcal{V}, \mathcal{E}_{global})$ is constructed, where $\mathcal{E}_{global} := \bigcup_{k \in \mathcal{K}} \mathcal{E}_k$ integrates all behavior interactions into a global multi-behavior heterogeneous graph. Then, Light-GCN (He et al., 2020) is used as the graph encoder:

$$\mathbf{E}_{global} = \text{LightGCN}(\mathcal{G}_{global}, \mathbf{E}^{(0)}), \qquad (7)$$

where $E^{(0)}$ is the randomly initialized user and item embeddings.

## 3.3. Auxiliary Behavior Structure Denoising Based on Graph Information Bottleneck

As shown in Figure 3, the overall objective of the proposed GCIB framework in the third part is to learn a denoised and information-compressed auxiliary behavior graph structure $\mathcal{G}'_k$ to enhance target behavior recommendation, rather than directly using the raw auxiliary behavior graph structure

$\mathcal{G}_k$. Due to the lack of prior knowledge for denoising, the target behavior user preference signal is introduced to guide the denoising of the auxiliary behavior graph. To balance the auxiliary behavior denoising and target behavior recommendation task, we optimize the GCIB model using the graph information bottleneck principle. Therefore, the objective for auxiliary behavior graph structure denoising is: Max : $I(\mathcal{R}; \mathcal{G}'_k) - \beta I(\mathcal{G}'_k; \mathcal{G}_k)$. The first term encourages the denoised auxiliary behavior graph to retain the essential information required for the target behavior recommendation task, which, according to prior work (Yang et al., 2024), can be equivalent to the BPR loss. The second term is the compression of the auxiliary behavior graph, aiming to filter out redundant auxiliary behavior information. Below, we describe the auxiliary behavior denoising guided by target behavior preference and the minimization of the mutual information between the denoised auxiliary behavior graph and the raw auxiliary behavior graph.

### • Target Behavior Preference Guided Auxiliary Behavior Denoising

To achieve the auxiliary behavior structure denoising objective mentioned above, the challenge is that although auxiliary behavior graphs contain noisy relationships, no labels guide the denoising process. To remove irrelevant information from the auxiliary behavior graph related to the target behavior recommendation, we propose to inject user target behavior preference signals into the denoising process.

Specifically, the auxiliary behavior graph denoising process is formulated as an edge drop problem. Given the original auxiliary behavior graph structure $\mathcal{G}_k$, the denoised auxiliary behavior graph is defined as:

$$\mathcal{G}'_k = \mathcal{F}_\phi(U, I, \mathcal{G}_k) = \{\mathcal{V}, \{e_{<u_a, i_b>} \odot \rho_{ab} \mid e_{<u_a, i_b>} \in \mathcal{E}_k\}\}, \qquad (8)$$

where $\rho_{ab} \sim \text{Bernoulli}(w_{ab})$ denotes that each edge $e_{<u_a, i_b>}$ will be dropped with probability $1 - w_{ab}$. Since there is no prior information, we introduce task-related tar-

get behavior user preferences and item representations to optimize the auxiliary behavior structure. Let $\mathbf{E}_{target}^U \in \mathbb{R}^{M \times d}$ and $\mathbf{E}_{target}^I \in \mathbb{R}^{N \times d}$ represent the learned user preference representation and item matrix from the target behavior interactions:

$$\mathbf{E}_{target} = \text{LightGCN}(\mathcal{G}_{target}, \mathbf{E}_{global}), \qquad (9)$$

For each observed auxiliary behavior relationship edge $e_{<u_a, i_b>}$, the edge confidence is computed as:

$$w_{ab} = f([\mathbf{e}_a; \mathbf{e}_b]), \qquad (10)$$

where $\mathbf{e}_a$ and $\mathbf{e}_b$ represent the learned user $a$ and item $b$ representations from target behavior interactions. $f(\cdot)$ is a fusion function, implemented as a one-layer MLP. The confidence score $w_{ab}$ can be interpreted as the target-aware retention probability of the auxiliary edge $e_{<u_a, i_b>}$. A larger $w_{ab}$ indicates that the auxiliary interaction is more consistent with the target-behavior preference, and is therefore more likely to be preserved in $\mathcal{G}_k'$. Since $\mathbf{E}_{target}$ is initialized from the global multi-behavior representation and further refined on the target graph, the confidence estimation uses both global behavioral context and target-specific supervision. However, the Bernoulli distribution parameter $\rho$ is non-differentiable, so we use the popular Concrete relaxation method (Jang et al., 2017) to replace it:

$$\text{Bernoulli}(w_{ab}) = \text{sigmoid}((\log(\delta/(1-\delta)) + w_{ab})/t)), \qquad (11)$$

where $\delta \sim U(0, 1)$, and $t \in \mathbb{R}^+$ is the temperature parameter. After reparameterization, the discrete Bernoulli distribution is replaced with a differentiable function.

● **Minimizing Mutual Information between Denoised and Original Auxiliary Behavior Graphs**

Next, we describe how to minimize $I(\mathcal{G}_k'; \mathcal{G}_k)$, which aims to reduce the redundant auxiliary behavior relationships in the original graph. Estimating the upper bound of mutual information is a challenging task. While some works have used variational techniques to estimate the upper bound (Alemi et al., 2017; Cheng et al., 2020), they heavily rely on prior assumptions. Thus, in this work, we follow prior work (Yang et al., 2024) and introduce the HSIC as an approximation for minimizing $I(\mathcal{G}_k'; \mathcal{G}_k)$.

**HSIC-based Graph Bottleneck Learning:** Given the original and denoised auxiliary behavior graph structures $\mathcal{G}_k$ and $\mathcal{G}_k'$, we minimize HSIC $(\mathcal{G}_k'; \mathcal{G}_k)$ as an approximation for minimizing $I(\mathcal{G}_k'; \mathcal{G}_k)$. However, graph structures are non-Euclidean data, making dependency measurements challenging. In practice, we use Monte Carlo sampling (Shapiro, 2003) to compute node representations for all nodes:

$$\text{Min: HSIC}(\mathcal{G}_k', \mathcal{G}_k) = H\hat{S}IC(\mathbf{E}_k'^{\mathbf{B}}, \mathbf{E}_k^{\mathbf{B}}), \qquad (12)$$

where $\mathbf{B}$ denotes the batch-sampled user and item nodes, and $\mathbf{E}_k'$ and $\mathbf{E}_k$ represent the denoised and original auxiliary behavior graph node representations, respectively. These representations are learned from the denoised model $\mathcal{G}_{\theta,\phi}(U, I, \mathcal{G}_k')$ and the original model $\mathcal{G}_\theta(U, I, \mathcal{G}_k)$. Thus, we reduce redundant auxiliary behavior relations through HSIC-based bottleneck regularization:

$$\mathcal{L}_{IB} = \frac{1}{|\mathcal{K}|} \sum_{k \in \mathcal{K}} H\hat{S}IC(\mathbf{E}_k'^{\mathbf{B}}, \mathbf{E}_k^{\mathbf{B}}). \qquad (13)$$

### 3.4. Cross-Behavior Semantic Alignment Based on Graph Contrastive Learning

The cross-behavior semantic alignment module aims to establish a semantic mapping bridge between the denoised auxiliary behavior graph and the target behavior graph through contrastive learning. This module leverages the rich information from denoised auxiliary behavior to compensate for the lack of supervised signals in target behavior data. The module first applies a graph information bottleneck mechanism to denoise the auxiliary behavior graph, followed by multi-layer graph convolution encoding to propagate node features and extract auxiliary behavior representations with high purity semantic features. Then, contrastive learning is used to achieve semantic alignment between the auxiliary behavior and target behavior.

The contrastive module complements the graph information bottleneck: while GIB filters task-irrelevant auxiliary structures, contrastive learning transfers useful semantics from the denoised auxiliary representations to the sparse target-behavior representations. The alignment is performed after structural denoising, so it aims to capture target-relevant common factors rather than forcing different behavior types to share identical semantics.

● **Target Behavior Domain Representation Learning**

Based on prior work on graph-based recommendation (Wu et al., 2021), we choose LightGCN as the embedding propagation encoder on the target behavior graph $\mathcal{G}_{target}$. For each node (user node as an example), the graph convolution at the $l^{th}$ layer is defined as:

$$u_{target}^{(l)} = \sum_{i \in \mathcal{N}_{\mathcal{G}_{target}(u)}} \frac{1}{\sqrt{|\mathcal{N}_{\mathcal{G}_{target}(u)}|}\sqrt{|\mathcal{N}_{\mathcal{G}_{target}(i)}|}} i^{(l-1)}, \qquad (14)$$

Similarly, for item nodes:

$$i_{target}^{(l)} = \sum_{u \in \mathcal{N}_{\mathcal{G}_{target}(i)}} \frac{1}{\sqrt{|\mathcal{N}_{\mathcal{G}_{target}(i)}|}\sqrt{|\mathcal{N}_{\mathcal{G}_{target}(u)}|}} u^{(l-1)}, \qquad (15)$$

● **Auxiliary Behavior Domain Representation Learning**

After obtaining the denoised auxiliary behavior graph $\mathcal{G}_k'$, we choose LightGCN as the embedding propagation encoder

on the denoised auxiliary behavior graph $\mathcal{G}'_k$. For each node (user node as an example), the graph convolution at the $l^{th}$ layer is defined as:

$$u_k^{(l)} = \sum_{i \in \mathcal{N}_{\mathcal{G}'_k}(u)} \frac{1}{\sqrt{|\mathcal{N}_{\mathcal{G}'_k}(u)|}\sqrt{|\mathcal{N}_{\mathcal{G}'_k}(i)|}} i_k^{(l-1)}, \quad (16)$$

Similarly, for item nodes:

$$i_k^{(l)} = \sum_{u \in \mathcal{N}_{\mathcal{G}'_k}(i)} \frac{1}{\sqrt{|\mathcal{N}_{\mathcal{G}'_k}(i)|}\sqrt{|\mathcal{N}_{\mathcal{G}'_k}(u)|}} u_k^{(l-1)}. \quad (17)$$

• **Cross-Behavior Semantic Alignment**

To establish semantic consistency between the target behavior domain and the auxiliary behavior domain and mitigate the lack of target behavior supervision signals, we design a cross-behavior alignment mechanism based on contrastive learning. Given user $u$ in the target behavior graph $\mathcal{G}_{\text{target}}$, its target behavior representation is $\mathbf{z}_u^{tgt} = \text{Mean}(u_{target}^{(0)}, \ldots, u_{target}^{(L_M-1)})$. Given user $u$'s denoised auxiliary behavior graph $\mathcal{G}_k$, its representation on the $k^{th}$ auxiliary behavior is $\mathbf{z}_u^{aux_k} = \text{Mean}(u_k^{(0)}, \ldots, u_k^{(L_M-1)})$. Finally, the auxiliary behavior representation for user $u$ is $\mathbf{z}_u^{aux} = \text{Mean}(\mathbf{z}_u^{aux_1}, \ldots, \mathbf{z}_u^{aux_k})$. To build the cross-behavior contrastive learning objective, we adopt the InfoNCE (Gutmann & Hyvärinen, 2010) loss to maximize the semantic mutual information between positive samples of target behavior and auxiliary behavior:

$$\mathcal{L}_{CL}^u = -\log \frac{\exp(s(\mathbf{z}_u^{tgt}, \mathbf{z}_u^{aux})/\tau)}{\sum_{u' \in \mathcal{N}_u^- \cup u} \exp(s(\mathbf{z}_u^{tgt}, \mathbf{z}_{u'}^{aux})/\tau)}, \quad (18)$$

where $s(\cdot)$ is the cosine similarity function, and $\tau$ is the contrastive temperature coefficient. Defining the item-side contrastive loss $\mathcal{L}_{CL}^i$ similarly, the final contrastive loss is:

$$\mathcal{L}_{CL} = \frac{1}{2}(\mathcal{L}_{CL}^u + \mathcal{L}_{CL}^i). \quad (19)$$

### 3.5. Model Prediction

The final model obtains user and item representations for the target behavior domain $\mathbf{z}_u^{tgt}, \mathbf{z}_i^{tgt}$ and for the denoised auxiliary behavior domain $\mathbf{z}_u^{aux}, \mathbf{z}_i^{aux}$. The target domain representation is more specific and detailed, while the auxiliary domain representation is more general. When the target behavior is sparse, the target domain representation may be less accurate. Therefore, we combine both representations:

$$e_u = \frac{1}{2}\left(\mathbf{z}_u^{tgt} + \mathbf{z}_u^{aux}\right), \quad (20)$$

$$e_i = \frac{1}{2}\left(\mathbf{z}_i^{tgt} + \mathbf{z}_i^{aux}\right). \quad (21)$$

Following (Mnih & Salakhutdinov, 2007), we use the inner product of the embeddings to predict recommendation score:

$$\hat{y}_{ui} = e_u \cdot e_i^\top. \quad (22)$$

*Table 1.* Statistics of four datasets

| Dataset | # User | # Item | # Interaction | Behavior Type |
|---|---|---|---|---|
| Tmall | 41,738 | 11,953 | $2.3 \times 10^6$ | {click, collect, cart, purchase} |
| Taobao | 48,749 | 39,493 | $2.0 \times 10^6$ | {click, cart, purchase} |
| Yelp | 19,800 | 22,734 | $1.4 \times 10^6$ | {tips, dislike, neutral, like} |
| ML-10M | 67,788 | 8,704 | $9.9 \times 10^6$ | {dislike, neutral, like} |

### 3.6. Model Optimization

To optimize the parameters of GCIB, we employ a multi-task training approach, optimizing both structural and semantic parts. Specifically, the structural part consists of the recommendation loss and graph information bottleneck loss, and the semantic part consists of the contrastive learning loss:

$$\mathcal{L} = \mathcal{L}_{BPR} + \beta \cdot \mathcal{L}_{IB} + \lambda \cdot \mathcal{L}_{CL} + \gamma \cdot \|\Theta\|_2. \quad (23)$$

Here, $\mathcal{L}_{BPR}$ is the target-behavior recommendation loss, which optimizes pairwise ranking over observed target interactions and sampled negative items. It serves as the task-predictive component of the information bottleneck objective. $\mathcal{L}_{IB}$ corresponds to the HSIC-based compression regularizer for auxiliary graph denoising, and $\mathcal{L}_{CL}$ encourages semantic alignment between denoised auxiliary representations and target-behavior representations.

## 4. Experiments

### 4.1. Datasets and Baselines

To evaluate the performance of GCIB[1], we conduct experiments on four public datasets: Tmall, Taobao, Yelp, and ML-10M. These datasets are commonly used multi-behavior recommendation benchmarks. The detailed statistics of all datasets are summarized in Table 1 and Appendix C.1. To assess the recommendation performance, we adopt two widely used metrics: Hit Ratio@K (*HR@K*) and Normalized Discounted Cumulative Gain@K (*NDCG@K*). *HR@K* measures whether the ground-truth item appears in the top-K recommendation list, while *NDCG@K* takes the rank position of the item into account, reflecting the quality of the ranking. We adopt the leave-one-out evaluation protocol (Jin et al., 2020) for all experiments.

We compare our proposed GCIB model with two categories of recommendation methods: (1) **Single-behavior methods**: MF-BPR (Loni et al., 2016) and LightGCN (He et al., 2020); (2) **Multi-behavior methods**: R-GCN (Schlichtkrull et al., 2018), NMTR (Gao et al., 2019), MBGCN (Jin et al., 2020), S-MBRec (Gu et al., 2022), CRGCN (Yan et al., 2023), MB-CGCN (Cheng et al., 2023), PKEF (Meng et al., 2023a), BCIPM (Yan et al., 2024), NSED (Cai et al., 2025) and MBLFE (Yan et al., 2025). Details of the baselines and experimental settings are provided in Appendix C.2 and C.3.

---

[1]The source code is available at: https://github.com/akajinchen/GCIB

*Table 2.* Performance comparison of GCIB with different models across multiple datasets(HR@10 and NDCG@10)

| Model | Tmall | | Taobao | | Yelp | | ML-10M | |
|---|---|---|---|---|---|---|---|---|
| | HR@10 | NDCG@10 | HR@10 | NDCG@10 | HR@10 | NDCG@10 | HR@10 | NDCG@10 |
| MF-BPR | 0.0236 | 0.0125 | 0.0173 | 0.0099 | 0.0320 | 0.0156 | 0.0579 | 0.0279 |
| LightGCN | 0.0383 | 0.0206 | 0.0261 | 0.0141 | 0.0393 | 0.0200 | 0.0669 | 0.0321 |
| R-GCN | 0.0309 | 0.0160 | 0.0282 | 0.0150 | 0.0347 | 0.0174 | 0.0552 | 0.0259 |
| NMTR | 0.0506 | 0.0249 | 0.0407 | 0.0213 | 0.0331 | 0.0160 | 0.0442 | 0.0197 |
| MBGCN | 0.0540 | 0.0277 | 0.0438 | 0.0266 | 0.0349 | 0.0185 | 0.0482 | 0.0233 |
| S-MBRec | 0.0708 | 0.0371 | 0.0572 | 0.0331 | 0.0351 | 0.0173 | 0.0331 | 0.0163 |
| CRGCN | 0.0837 | 0.0432 | 0.1126 | 0.0620 | 0.0370 | 0.0179 | 0.0504 | 0.0243 |
| MB-CGCN | 0.1102 | 0.0406 | 0.0973 | 0.0464 | 0.0348 | 0.0163 | 0.0644 | 0.0306 |
| PKEF | 0.1100 | 0.0644 | 0.1086 | 0.0610 | 0.0426 | 0.0213 | 0.0583 | 0.0265 |
| BCIPM | 0.1445 | 0.0831 | 0.1428 | 0.0817 | 0.0502 | 0.0244 | 0.0810 | 0.0392 |
| NSED | 0.1502 | 0.0810 | 0.1416 | 0.1004 | 0.0355 | 0.0146 | 0.0377 | 0.0162 |
| MBLFE | 0.1291 | 0.0696 | 0.1577 | 0.0907 | 0.0531 | 0.0261 | 0.0659 | 0.0310 |
| **GCIB(Ours)** | **0.1617** | **0.0944** | **0.1815** | **0.1199** | **0.0746** | **0.0358** | **0.0916** | **0.0429** |
| **Impr.** | **+7.66%** | **+13.60%** | **+15.09%** | **+19.42%** | **+40.49%** | **+37.16%** | **+13.09%** | **+9.44%** |

*Table 3.* Performance comparison of GCIB with different models across multiple datasets (HR@20 and NDCG@20)

| Model | Tmall | | Taobao | | Yelp | | ML-10M | |
|---|---|---|---|---|---|---|---|---|
| | HR@20 | NDCG@20 | HR@20 | NDCG@20 | HR@20 | NDCG@20 | HR@20 | NDCG@20 |
| MF-BPR | 0.0285 | 0.0140 | 0.0210 | 0.0119 | 0.0377 | 0.0176 | 0.0793 | 0.0326 |
| LightGCN | 0.0482 | 0.0233 | 0.0344 | 0.0165 | 0.0474 | 0.0217 | 0.0904 | 0.0344 |
| R-GCN | 0.0413 | 0.0181 | 0.0326 | 0.0169 | 0.0423 | 0.0195 | 0.0695 | 0.0297 |
| NMTR | 0.0675 | 0.0292 | 0.0556 | 0.0231 | 0.0384 | 0.0189 | 0.0534 | 0.0219 |
| MBGCN | 0.0699 | 0.0294 | 0.0562 | 0.0287 | 0.0449 | 0.0218 | 0.0601 | 0.0265 |
| S-MBRec | 0.0827 | 0.0444 | 0.0720 | 0.0348 | 0.0440 | 0.0183 | 0.0427 | 0.0193 |
| CRGCN | 0.1059 | 0.0493 | 0.1535 | 0.0726 | 0.0510 | 0.0199 | 0.0690 | 0.0272 |
| MB-CGCN | 0.1409 | 0.0451 | 0.1281 | 0.0538 | 0.0486 | 0.0174 | 0.0878 | 0.0364 |
| PKEF | 0.1351 | 0.0718 | 0.1500 | 0.0716 | 0.0502 | 0.0243 | 0.0761 | 0.0288 |
| BCIPM | 0.1999 | 0.0970 | 0.1952 | 0.0949 | 0.0874 | 0.0377 | 0.1433 | 0.0548 |
| NSED | 0.2042 | 0.0943 | 0.1569 | 0.1043 | 0.0635 | 0.0216 | 0.0714 | 0.0247 |
| MBLFE | 0.1865 | 0.0836 | 0.2144 | 0.1049 | 0.0896 | 0.0352 | 0.1123 | 0.0427 |
| **GCIB(Ours)** | **0.2200** | **0.1095** | **0.2198** | **0.1296** | **0.1180** | **0.0468** | **0.1610** | **0.0611** |
| **Impr.** | **+7.74%** | **+12.89%** | **+2.52%** | **+23.55%** | **+31.70%** | **+24.14%** | **+12.35%** | **+11.50%** |

## 4.2. Overall Performance Analysis

We compare GCIB with all baseline methods, and the overall results on four benchmark datasets (Tmall, Taobao, Yelp, and ML-10M) are summarized in Table 2 and Table 3. Performance is evaluated using *HR@10*, *NDCG@10*, *HR@20*, and *NDCG@20*, where the best and second-best results are highlighted in bold and underlined.

Overall, GCIB consistently outperforms all baselines across all datasets and evaluation metrics, demonstrating the effectiveness of its graph bottleneck-based denoising and contrastive alignment strategies. Compared to the best baseline, GCIB achieves significant improvements—up to 40.5% on the sparse Yelp dataset and around 13.1% on the dense ML-10M dataset—showing strong adaptability under varying data sparsity conditions. These results highlight that GCIB effectively filters out noisy auxiliary behaviors through graph information bottleneck and enhances target behav-

ior representations via cross-behavior semantic alignment.

Moreover, GCIB alleviates the negative transfer problem commonly observed in cascaded multi-behavior models such as CRGCN and MB-CGCN by jointly optimizing structural and semantic dependencies. Compared with recent denoising or self-supervised baselines such as BCIPM, NSED, and MBLFE, GCIB further benefits from explicitly coupling structure-level edge filtering with feature-level semantic alignment. The structural-semantic co-optimization design allows GCIB to maintain stable performance in both sparse and noisy environments, demonstrating its superior generalization capability and robustness.

## 4.3. Ablation Study

To further examine the effectiveness of each GCIB component, we perform ablation experiments on the Tmall, Taobao, and Yelp datasets, with results summarized in Table 4.

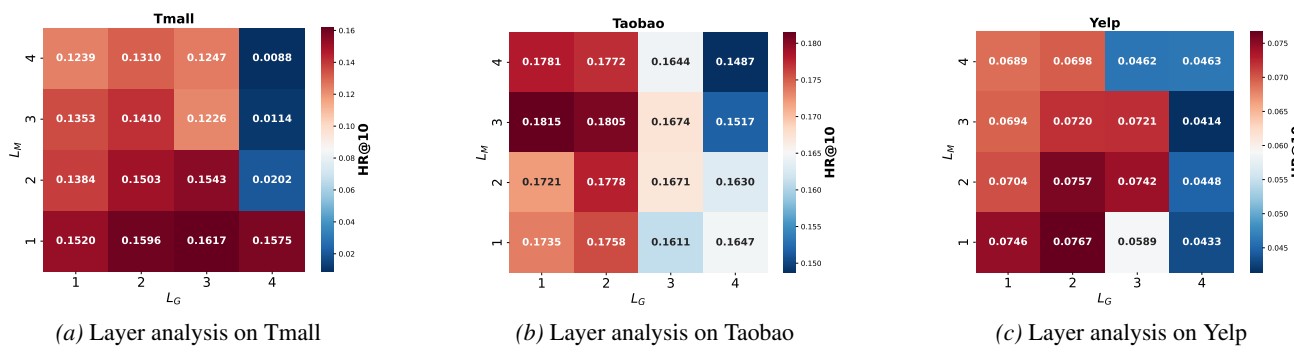

*(a)* Layer analysis on Tmall  *(b)* Layer analysis on Taobao  *(c)* Layer analysis on Yelp

*Figure 4.* Analysis of global encoding layers $L_G$ and auxiliary/target behavior layers $L_M$ across multiple datasets

*Table 4.* Ablation study results on different datasets

| Dataset | Metric | -Global | -IB | -Infonce | -Both | **GCIB(Ours)** |
|---------|--------|---------|-----|----------|-------|----------------|
| Tmall | HR@10 | 0.1101 | 0.1089 | 0.1523 | 0.0356 | **0.1617** |
| | NDCG@10 | 0.0669 | 0.0618 | 0.0906 | 0.0190 | **0.0944** |
| Taobao | HR@10 | 0.1666 | 0.1724 | 0.1661 | 0.0352 | **0.1815** |
| | NDCG@10 | 0.1118 | 0.1138 | 0.1080 | 0.0202 | **0.1199** |
| Yelp | HR@10 | 0.0698 | 0.0700 | 0.0641 | 0.0522 | **0.0746** |
| | NDCG@10 | 0.0351 | 0.0348 | 0.0317 | 0.0262 | **0.0358** |

• **Global Encoding on Heterogeneous Multi-Behavior Graph.** Removing the global encoding module (*-Global*) leads to a consistent decline across all datasets, confirming that the unified multi-behavior graph provides a useful initialization before behavior-specific propagation. This global context helps capture cross-behavior collaborative patterns and stabilizes the subsequent target and auxiliary representation learning.

• **Information Bottleneck-based Auxiliary Behavior Denoising.** Excluding the denoising mechanism (*-IB*) results in clear performance degradation, especially on Tmall, indicating that noisy or task-irrelevant auxiliary interactions can weaken target-behavior prediction if directly propagated. This verifies the necessity of structure-level auxiliary graph filtering before representation learning.

• **Cross-Behavior Semantic Alignment via Contrastive Learning.** When the contrastive alignment module is removed (*-Infonce*), performance drops on all three datasets, with more evident decreases on Taobao and Yelp. This shows that cross-behavior contrastive learning complements structural denoising by transferring useful semantics from denoised auxiliary representations to sparse target-behavior representations.

• **Structure–Semantics Dual-Domain Optimization.** The variant without both denoising and alignment components (*-Both*) exhibits the largest drop in performance, verifying that structural refinement and semantic alignment are both important. The former suppresses noisy auxiliary structures, while the latter enriches target representations with comple-

mentary auxiliary semantics, and their combination yields more robust representation learning.

Overall, the ablation results validate that each component contributes to GCIB's effectiveness, and that combining graph-based structural refinement with semantic-level alignment yields the most substantial performance gains.

### 4.4. Structural Robustness Analysis

To better understand how the proposed IB-based denoising mechanism affects auxiliary graph structures, we further analyze its edge retention behavior and robustness to injected auxiliary noise.

**Edge Retention.** We conduct a post-hoc analysis on the retained auxiliary edges after denoising. On Tmall, the cart and click views retain 70.94% and 60.42% of their edges, respectively, indicating that GCIB performs behavior-aware filtering rather than uniform pruning. The lower retention ratio of the denser click view suggests that dense auxiliary behaviors may contain more generic or task-irrelevant interactions and thus require stronger filtering.

**Auxiliary Noise Robustness.** We then conduct a controlled noise injection experiment on Yelp. Specifically, we inject 20% synthetic fake interactions into the tip auxiliary graph by randomly sampling unobserved user–item pairs. As shown in Table 5, GCIB remains stable after noise injection, while MBLFE (Yan et al., 2025) and HGIB (Zhang et al., 2025) suffer more substantial performance degradation. This result provides direct evidence that GCIB can selectively filter task-irrelevant auxiliary edges and maintain an effective graph structure under perturbations.

Overall, these analyses show that GCIB performs structure-aware and selective edge filtering rather than indiscriminate pruning, and is robust to noisy auxiliary interactions.

### 4.5. Hyperparameter Analysis

We analyze the sensitivity of the key hyperparameters in GCIB, including the information bottleneck loss weight $\beta$,

*Table 5.* Robustness to synthetic auxiliary noise on Yelp.

| Method | Setting | HR@10 | NDCG@10 | HR@20 | NDCG@20 |
|---|---|---|---|---|---|
| GCIB | Clean | 0.0746 | 0.0358 | 0.1180 | 0.0468 |
| | + Noise | 0.0719 | 0.0372 | 0.1169 | 0.0484 |
| | Rel. Change | -3.62% | +3.91% | -0.93% | +3.42% |
| MBLFE | Clean | 0.0531 | 0.0261 | 0.0896 | 0.0352 |
| | + Noise | 0.0429 | 0.0213 | 0.0719 | 0.0286 |
| | Rel. Change | -19.21% | -18.39% | -19.75% | -18.75% |
| HGIB | Clean | 0.0352 | 0.0139 | 0.0711 | 0.0228 |
| | + Noise | 0.0312 | 0.0127 | 0.0690 | 0.0221 |
| | Rel. Change | -11.36% | -8.63% | -2.95% | -3.07% |

the contrastive loss weight $\lambda$, the number of global encoding layers $L_G$, and the number of layers for auxiliary and target behavior graphs $L_M$. Experiments are conducted on the Tmall, Taobao, and Yelp datasets.

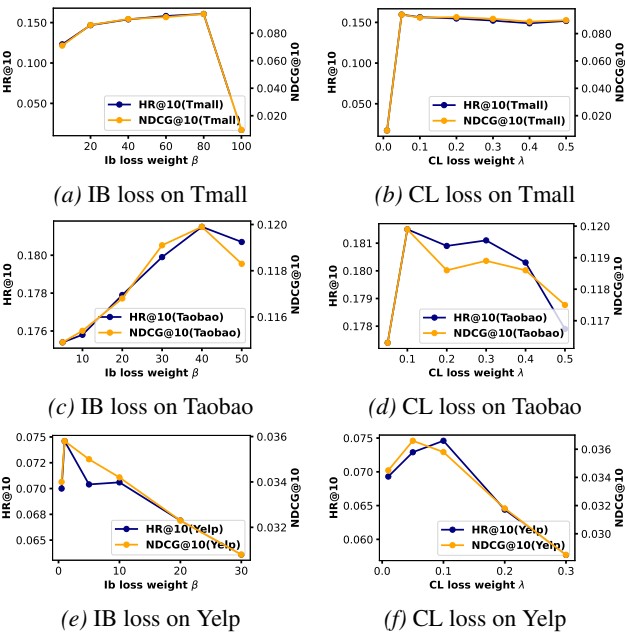

*(a)* IB loss on Tmall     *(b)* CL loss on Tmall

*(c)* IB loss on Taobao     *(d)* CL loss on Taobao

*(e)* IB loss on Yelp     *(f)* CL loss on Yelp

*Figure 5.* Analysis of loss weights $\beta$ and $\lambda$. (a)-(b) Tmall; (c)-(d) Taobao; (e)-(f) Yelp.

• **Information Bottleneck Loss Weight** $\beta$**.** As shown in Figures 5a, 5c, and 5e, we examine the impact of $\beta$ on Tmall, Taobao, and Yelp. The performance of GCIB first increases as $\beta$ grows, then decreases when $\beta$ becomes too large. An appropriate value of $\beta$ helps the model extract more effective denoised signals from auxiliary behavior graphs to support the target recommendation task. In contrast, inappropriate values may lead to misaligned auxiliary information and reduce performance. Specifically, GCIB achieves its best performance with $\beta = 80$ on Tmall, $\beta = 40$ on Taobao, and $\beta = 1$ on Yelp. The difference in the optimal $\beta$ is mainly data-dependent, as datasets vary in sparsity, behavior semantics, noise levels, and auxiliary–target behavior relationships.

Moreover, the HSIC-based IB term is numerically small, so a relatively larger coefficient may be needed on some datasets to provide effective regularization, while overly large values can over-compress useful auxiliary signals.

• **Contrastive Loss Weight** $\lambda$**.** Parameter $\lambda$ controls the magnitude of contrastive learning loss. We vary $\lambda$ from 0.01 to 0.5. In Figures 5b, 5d, and 5f, the model performance improves as $\lambda$ increases up to a threshold, suggesting that a moderate contrastive weight effectively aligns the semantics between target and denoised auxiliary behaviors, enhancing robustness and recommendation accuracy. When $\lambda$ is too small, the auxiliary semantic signals cannot be sufficiently transferred to the sparse target behavior representations. However, overly large $\lambda$ may make the contrastive objective dominate the recommendation objective and force excessive cross-behavior alignment, thereby impairing the final representation quality.

• **Global Encoding Layers** $L_G$ **and Behavior Graph Layers** $L_M$**.** As shown in Figures 4a, 4b, and 4c, we vary the number of layers from 1 to 4. Results reveal that the optimal layer configuration differs across datasets. In general, shallow global encoding ($L_G = 2$ or 3) achieves better performance, while deeper $L_G$ tends to cause overfitting. Likewise, appropriate $L_M$ improves recommendation quality since the information bottleneck can refine auxiliary graph signals. However, excessively deep $L_M$ may introduce redundant message passing, leading to over-smoothing and noise accumulation in multi-behavior graphs, thereby degrading performance.

## 5. Conclusion

In this paper, we introduce GCIB (**G**raph **C**ontrastive **I**nformation **B**ottleneck), a novel framework for multi-behavior recommendation that addresses the challenge of noisy or irrelevant auxiliary behaviors under sparse target interactions. GCIB employs a dual-level denoising strategy: at the graph structural level, it leverages the information bottleneck principle to prune spurious auxiliary connections and preserve only task-relevant interaction patterns; at the feature level, it applies a cross-behavior contrastive learning scheme to align and enrich target behavior representations with complementary semantics distilled from auxiliary data. Through this principled design, our approach effectively filters out noise and enhances target representations, enabling noise-resilient, target-aware user and item embeddings for recommendation. Extensive experiments on multiple real-world datasets validate the efficacy of GCIB, as it consistently outperforms state-of-the-art baselines and demonstrates robust performance across diverse scenarios. In future work, we plan to extend GCIB by exploring dynamic behavior modeling and graph-based generative augmentation strategies, which could further enhance its adaptability.

## Acknowledgements

We would like to declare that Likang Wu and Zihao Chen contributed equally to this work and should be considered co-first authors. This work was supported by the National Natural Science Foundation of China (Nos. 62502340, 61976001), the Natural Science Foundation of Anhui Province (Nos. 2508085MF176, 2408085MF152), the Natural Science Foundation of Tianjin (No. 24JCQNJC01560), and also supported by the Key Projects of University Excellent Talents Support Plan of Anhui Provincial Department of Education (No. gxyqZD2021089).

## Impact Statement

This paper aims to advance Machine Learning by introducing a graph contrastive information bottleneck framework for robust multi-behavior recommendation. The contribution is primarily methodological, focusing on denoising auxiliary behaviors and improving representation learning under sparse target interactions. While the work does not involve direct deployment or sensitive user data, recommendation systems may influence user choices and item exposure in practical applications. Therefore, future use of this method should consider fairness, privacy protection, and the potential impact on long-tail users and items.

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

# A. Related Work

**Graph-based Recommendation.** Collaborative filtering has long been a foundational technique in recommendation systems. Traditional approaches, such as matrix factorization (Zhao et al., 2015), aim to learn user and item latent representations from observed interactions. With the advent of graph neural networks, the field has witnessed significant progress by representing user-item interactions as bipartite graphs and propagating signals through graph structures to capture higher-order collaborative information. One of the early graph-based models, NGCF (Wang et al., 2019c), applies non-linear message passing over the user-item graph, effectively modeling both direct and high-order connections. Subsequently, LightGCN (He et al., 2020) streamlines this architecture by removing feature transformation and activation functions, focusing on weighted neighborhood aggregation, which greatly improves scalability and effectiveness. Building upon these backbones, various efforts have further enhanced GNN-based recommenders. PinSage (Ying et al., 2018) introduces random walks and neighbor sampling to handle large-scale graphs efficiently. SGL(Wu et al., 2021) incorporates self-supervised learning to reduce reliance on labeled data and enhance robustness against data sparsity. Our work continues in this direction by proposing a principled framework to address structural noise and enhance representation quality in multi-behavior recommendation.

**Multi-Behavior Recommendation.** Multi-behavior recommendation aims to alleviate data sparsity by leveraging multiple user-item interactions, like clicks, collections, and purchases, to enhance the prediction of target behaviors. Early approaches primarily employed matrix factorization (MF) techniques to handle multi-behavior data. For example, Collaborative Matrix Factorization (CMF) (Zhao et al., 2015) utilized a shared embedding space to model different behavior types. MF-BPR (Loni et al., 2016) introduced and refined negative sampling strategies to enrich the training samples, further improving model performance. Subsequent works by researchers have explored multi - behavior recommendation models based on neural and graph architectures. DNN-based methods typically learn embeddings for each behavior and integrate them for target behavior prediction. For instance, NMTR (Gao et al., 2019) models cascading dependencies by passing predicted scores of earlier behaviors to later ones. GCN-based models, on the other hand, construct unified interaction graphs to capture higher-order user–item relationships. Representative methods such as MBGCN (Jin et al., 2020) and CRGCN (Yan et al., 2023) exploit hierarchical dependencies among behaviors to refine user preference representations. However, these cascaded models often suffer from imbalanced or noisy auxiliary behaviors, leading to negative transfer. To address this, PKEF (Meng et al., 2023a) introduces a hybrid architecture combining parallel and cascaded learning paradigms to mitigate noise and sparsity issues.

Furthermore, self-supervised learning (SSL) techniques have been explored to improve semantic alignment between target and auxiliary behaviors. For example, S-MBRec (Gu et al., 2022) employs star-style contrastive learning to capture commonalities between their embeddings, alleviating supervision sparsity and redundancy; BCIPM (Yan et al., 2024) focuses on denoising by learning behavior-contextualized preferences and retaining only target-relevant information, with pre-training to enhance sparse scenario performance. While these SSL-inspired methods advance multi-behavior recommendation in structured scenarios, they still struggle with structural noise and semantic misalignment in auxiliary behaviors. Our work addresses these gaps by integrating Graph Information Bottleneck (GIB) for structural refinement and self-supervised contrastive learning for cross-behavior alignment—jointly enhancing useful signal extraction and noise mitigation to boost target behavior prediction accuracy.

**Graph Information Bottleneck.** The Information Bottleneck (IB) principle aims to mitigate noise and redundancy in data by preserving only the most relevant information. In graph-based recommendation systems, this translates to minimizing the mutual information between the input graph and its learned representations, while maximizing the mutual information between these representations and the recommendation task. Early works like InfoGraph (Sun et al., 2019) applied IB to enhance graph embedding expressiveness, laying the foundation for subsequent advancements that integrate IB with contrastive learning—such as CGI (Wei et al., 2022), which uses graph augmentation to improve representation robustness by leveraging contrastive signals.

To address noise from irrelevant graph connections, Graph Information Bottleneck (GIB) models extend the IB framework to structured graph data. These methods minimize mutual information with noisy graph structures while maximizing the extraction of task-useful information. IGCL (Guo et al., 2024) encourages the model to distinguish between similar and dissimilar graph structures, thereby enhancing the robustness of node representations and better filtering out structural noise. GBSR (Yang et al., 2024) and IBMRec (Yang et al., 2025) apply IB-based denoising to social graph data and multi-modal features, respectively; HGIB (Zhang et al., 2025) introduces a model-agnostic Hierarchical Graph Information Bottleneck framework for multi-behavior recommendation, addressing distribution disparities and negative transfer. Building

on this line of research, our work applies GIB to multi-behavior recommendation: we model auxiliary behavior graph structures and introduce a structural bottleneck to optimally filter noise while preserving task-relevant information for target behavior prediction. Different from these representation-level or modality-specific IB methods, GCIB applies the bottleneck directly to auxiliary behavior graph structures by learning target-guided edge retention masks before graph propagation. This structural formulation enables GCIB to suppress noisy auxiliary interactions at their source, rather than compressing representations after noisy messages have already been propagated.

*Table 6.* Training time, inference time, and GPU memory usage on Tmall, Taobao, and Yelp

| Models | Tmall | | | Taobao | | | Yelp | | |
|---|---|---|---|---|---|---|---|---|---|
| | time/epoch | infer time | GPU(MB) | time/epoch | infer time | GPU(MB) | time/epoch | infer time | GPU(MB) |
| LightGCN | 2.51s | 0.003s | 1302 | 2.33s | 0.003s | 1674 | 9.91s | 0.05s | 1504 |
| BCIPM | 495.57s | 4.63s | 2742 | 406.95s | 4.58s | 2976 | 912.81s | 6.39s | 4592 |
| **GCIB(Ours)** | 30.41s | 0.058s | 5218 | 27.772s | 0.059s | 4816 | 37.916s | 0.026s | 2306 |

## B. Model Complexity Analysis

**Space complexity**: Storing the embedding matrices for $n$ users and $m$ items in $d$-dimensional space requires $O((n+m) \times d)$ memory. The model also maintains adjacency lists for all observed interactions (total $|E|$ edges across behaviors), which adds $O(|E|)$ space. Any additional parameters from the masking networks and projection heads are minimal in size (e.g., small MLP weights) and thus negligible compared to the embedding storage.

**Time complexity**: Each training epoch entails the $L$-layer LightGCN propagation over the user-item interaction graph, which takes $O(L \times |E| \times d)$ operations as the dominant cost. The novel components of GCIB incur only minor linear-time overhead on top of this. The learned auxiliary-graph masking module involves simple edge-wise computations (sparse probability calculations for each observed auxiliary edge), scaling on the order of $|E|$ with efficient sparse matrix operations. The HSIC-based information bottleneck regularizer is computed on a small sampled subset of nodes, so its cost per batch is trivial (constant or linear in $d$) and does not significantly affect the overall complexity. Likewise, the cross-behavior InfoNCE alignment is implemented with effective negative sampling, keeping its complexity roughly linear in the number of nodes (users/items) involved. Combining all components, the total per-epoch time complexity of GCIB can be summarized as $O(L \times |E| \times d)$ (with only modest additional constants from the IB and contrastive modules). The overall computational complexity therefore remains comparable to standard GCN-based recommenders, scaling linearly with the number of users, items, and observed interactions.

## C. Additional Experimental Details

### C.1. Detailed Descriptions of Datasets

- **Tmall.** Tmall is another e-commerce platform owned by Alibaba Group. This dataset contains four types of user behaviors: click, collect, add-to-cart, and purchase. Among them, purchase is treated as the target behavior

- **Taobao.** This dataset is collected from the Taobao platform, an e-commerce site also operated by Alibaba Group. It includes three types of user behaviors: click, add-to-cart, and purchase, where purchase serves as the target behavior.

- **Yelp.** This dataset is collected from Yelp, a popular online review platform in the U.S. It contains *tips* and *ratings*. The ratings $r$ are categorized into three behavior types: dislike ($r \leq 2$), neutral ($2 < r < 4$), and like ($r \geq 4$). The like behavior is treated as the target, while tips, dislike, and neutral are considered auxiliary behaviors.

- **ML-10M.** This dataset is provided by the GroupLens research group and widely used for movie recommendation. It includes user ratings on movies, where ratings $r$ are also divided into dislike ($r \leq 2$), neutral ($2 < r < 4$), and like ($r \geq 4$). The like behavior is regarded as the target behavior, and the rest as auxiliary behaviors.

### C.2. Detailed Descriptions of Baselines

**Single-behavior models. MF-BPR** (Loni et al., 2016) adopts matrix factorization with the Bayesian Personalized Ranking (BPR) loss to model pairwise item preferences. **LightGCN** (He et al., 2020) simplifies graph convolutional operations by removing nonlinear transformations, propagating embeddings directly along user–item edges.

**Multi-behavior models.** **R-GCN** (Schlichtkrull et al., 2018) captures heterogeneous relations with behavior-specific weight matrices. **NMTR** (Gao et al., 2019) models cascading dependencies across behaviors using multi-task learning. **MBGCN** (Jin et al., 2020) constructs behavior-specific user–item graphs and fuses them via importance weighting. **S-MBRec** (Gu et al., 2022) introduces star-style contrastive learning for cross-behavior alignment. **CRGCN** (Yan et al., 2023) uses cascading residual GCNs to refine embeddings along behavior sequences. **MB-CGCN** (Cheng et al., 2023) simplifies CRGCN with LightGCN propagation and feature transformation layers. **PKEF** (Meng et al., 2023a) incorporates parallel knowledge fusion and mixture-of-experts for balanced training. **BCIPM** (Yan et al., 2024) employs graph-based denoising and pretraining for high-order preference modeling. **NSED** (Cai et al., 2025) leverages neighborhood-enhanced GCNs and a denoising module for multi-behavior recommendation. **MBLFE** (Yan et al., 2025) employs a gating expert network with self-supervised factor disentanglement to model latent user preferences from multi-behavior data.

### C.3. Experimental Settings

We implement all models in PyTorch and DGL (Wang et al., 2019a). For evaluation, we adopt the widely-used leave-one-out protocol. We use the Adam optimizer (Kingma & Ba, 2015) with a learning rate of 0.001. The batch size is 1024 (4096 for ML-10M). We set the embedding dimension to 64. The number of global encoding layers $L_G$ is selected from $\{1, 2, 3, 4\}$, and the number of message-passing layers $L_M$ for both auxiliary and target behaviors is selected from $\{1, 2, 3, 4\}$. The information bottleneck loss weight $\beta$ is selected from [1, 100], and the contrastive loss weight $\lambda$ is selected from [0.01, 0.5]. The temperature parameter for InfoNCE loss is 0.05 for Tmall, 0.15 for Taobao, and 0.2 for Yelp and ML-10M. The parameter $\sigma$ of the RBF kernel is set to 0.25. Weight decay is selected from $\{0, 1e-5\}$. For all baselines, we follow tuning protocols reported in (Yan et al., 2024).

## D. Model Efficiency Analysis

Table 6 reports the training time, inference time, and GPU memory consumption of GCIB compared with several representative baselines. LightGCN, as a single-behavior model, has the lowest per-epoch training cost due to its lightweight structure, though it lacks the capacity to exploit auxiliary behaviors. Among the multi-behavior methods, GCIB demonstrates substantially lower computational overhead than BCIPM, which requires longer training time and higher inference latency. In terms of GPU memory usage, GCIB maintains a comparable level to other multi-behavior approaches, remaining within the practical range of modern GPUs. These results supplement the efficiency analysis in the main paper, confirming that GCIB achieves a favorable balance between model complexity and computational efficiency.

## E. Embedding Visualization Analysis

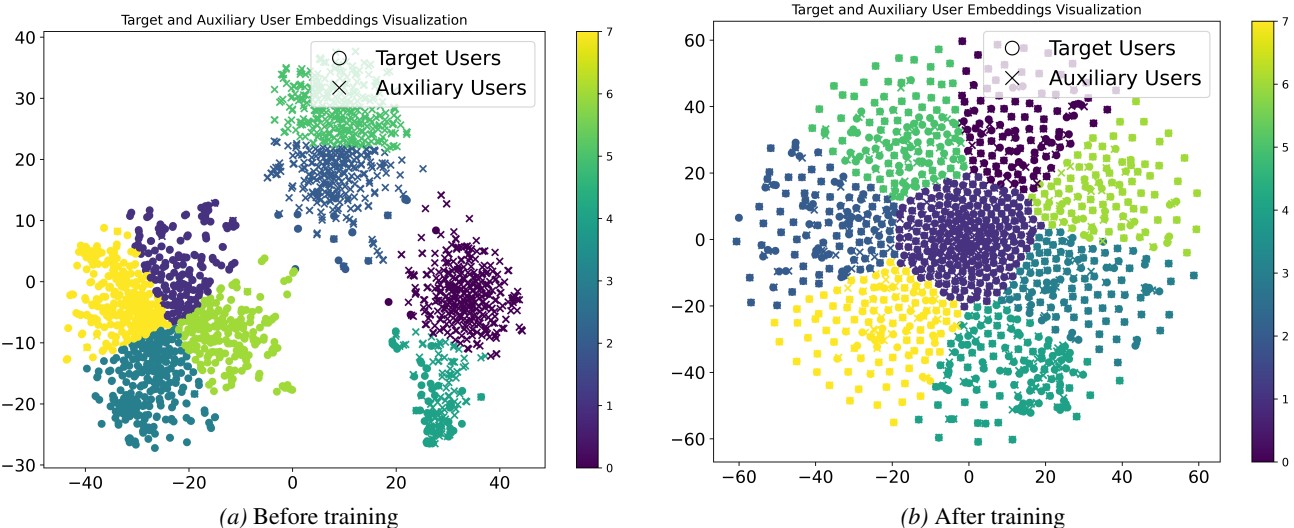

*(a)* Before training        *(b)* After training

*Figure 6.* Visualization of user embedding distributions on Tmall: (a) initial state vs. (b) after GCIB training.

This section provides supplementary visualization results to illustrate the embedding alignment between auxiliary and target

behaviors. A total of 1,000 users are randomly sampled from the Tmall dataset, and their embeddings from both domains are extracted at inference time. All embeddings are $\ell_2$-normalized, reduced to two dimensions using t-SNE, and clustered via $k$-means ($k$=8). Figure 6 presents user embedding distributions before and after training.

Before training, the auxiliary and target embeddings occupy distinct regions in the latent space, suggesting weak correlation across behavior domains. After training, the embeddings become more intermixed and form compact clusters, reflecting improved cross-behavior consistency. This phenomenon visually supports the alignment effects described in the main paper, where GCIB integrates structural denoising and semantic consistency to unify behavior representations.

