# OpenReview forum: "GCIB: Graph Contrastive Information Bottleneck for Multi-Behavior Recommendation"
_ICML.cc/2026/Conference — ICML 2026 regular_

### Official Review · Reviewer_np72 · 2026-03-06

**Soundness:** 2
**Presentation:** 3
**Significance:** 3
**Originality:** 3
**Overall Recommendation:** 4
**Confidence:** 4

**Summary:**

Auxiliary user behaviours can effectively alleviate data sparsity, however, they also introduce noise and irrelevant interactions. To address this issue, this paper proposes the graph contrastive information bottleneck, which denoises auxiliary behaviours at both structural and feature levels. Extensive experiments demonstrate GCIB outperforms state-of-the-art baselines.

**Compliance With Llm Reviewing Policy:**

Affirmed.

**Final Justification:**

Thanks so much for your carefully revised, considering of the comments form other reviewers. I maintain my recommendation.

**Key Questions For Authors:**

Please refer to the questions in Weakness and Limitation sections.

**Limitations:**

1. This paper aim to alleviate the problem of target data sparsity; however, the proposed target-guided denoising mechanism relies on the availability of accurate target embeddings to function effectively. In sparse regimes, particularly for long-tail and cold start users, learning accurate embeddings is precisely the core challenges. Consequently, the method assumes the solution it seeks to provide, creating a logical contradiction.
2. The bidirectional feedback loop of target and auxiliary embeddings may susceptible to a vicious cycle.
3. Please provide training loss curves or embedding similarity metrics to demonstrate convergence stability despite the circular dependency.

**Strengths And Weaknesses:**

Strength:
This paper proposes a novel framework that performs denoising from structure and feature representations. This paper conduct comprehensive experiments on multiple datasets, and the results demonstrate GCIB outperforms state-of-the-art baselines.

Weakness:
1. The interdependent relationships between target and auxiliary embeddings create a bidirectional feedback loop, which may susceptible to a vicious cycle, if erroneous signals are introduced at any stage, they can be progressively amplified through the interdependent updates, leading to cumulative error propagation rather than convergence.
2. The denoising mechanism relies on collaborative signals from target behaviours, which risks over smoothing user representations. long-tail and personalized preference might be pruned as noise.
3. The target-guided denoising mechanism relies on accurate target embedding, which may unavailable for cold-start users and unstable in early training epochs.

---

> ### Author Rebuttal · Authors · 2026-03-30
>
> We thank the reviewer for raising important practical concerns.
>
> **For W1**. We agree that GCIB involves interaction between the target and auxiliary branches during joint optimization, and thus the concern about possible error amplification is reasonable. However, we would like to clarify that this interaction does not constitute an explicit recursive feedback loop.
>
> Specifically, target-behavior embeddings are primarily learned from the target graph encoder and the target recommendation objective, while auxiliary behaviors are refined through IB-based edge filtering. The role of target-side representations in the denoising module is to provide a guidance signal for estimating edge confidence, rather than being repeatedly overwritten by auxiliary updates. Conversely, the auxiliary branch influences the target side mainly through the joint contrastive/recommendation objectives, which provide complementary supervision rather than recursively feeding back intermediate errors.
>
> Therefore, the dependency in GCIB is better characterized as weakly coupled joint optimization rather than a strong bidirectional loop that progressively amplifies errors. We will clarify this distinction in the revised manuscript and provide additional training-loss analysis to further demonstrate stable convergence in practice.
>
> **For W2.** We thank the reviewer for this insightful concern regarding potential over-smoothing and the risk of pruning long-tail preferences.
>
> To examine whether GCIB disproportionately removes sparse or personalized signals, we conduct a post-hoc analysis on Tmall by grouping auxiliary edges according to users’ target-behavior degrees (sparse / normal / dense) and comparing their retention behavior.
>
> In the auxiliary behavior view with the largest number of edges, sparse users exhibit a higher survival rate than dense users (65.83% vs. 50.24%). This indicates that GCIB does not indiscriminately prune low-frequency interactions. Instead, it tends to preserve edges associated with sparse users, while more aggressively filtering edges from dense users, which are more likely to contain redundant or noisy interactions. This observation suggests that the proposed denoising mechanism is adaptive rather than frequency-biased, and does not suppress long-tail preferences.
>
> In addition, we provide embedding visualization results in the appendix, which show that after training, user representations form more compact yet still distinguishable clusters, without collapsing into overly smooth distributions. This further indicates that GCIB avoids over-smoothing while improving representation consistency.
>
> **For W3**. We agree that extreme cold-start settings are challenging for any target-guided denoising strategy. In GCIB, however, the target-side embeddings are not learned from scratch only on the sparse target graph: they are initialized by the global multi-behavior encoder and then refined on the target graph. Therefore, the guidance signal is informed by both global multi-behavior structure and target-specific refinement. We will clarify this point and add a discussion of the extreme cold-start limitation in the revised paper.

---

> > ### Author Rebuttal · Reviewer_np72 · 2026-04-02
> >
> > Thanks so much for your carefully revised, which partially resolved my concerns. I also acknowledge that the concerns raised by reviewers RLFt and  CAQa are reasonable and should be carefully considered. I maintain my recommendation.

---

> > > ### Author Response · Authors · 2026-04-06
> > >
> > > We sincerely thank you for your positive evaluation and for recognizing the technical solidity of our work. Your highly insightful questions regarding the embedding learning dynamics—specifically the potential for error propagation, the risk of over-smoothing, and the dependency on target embeddings for cold-start users—have genuinely helped us improve the theoretical rigorousness of our paper.
> > >
> > > Thank you again for your time, expertise, and support!

---

### Official Review · Reviewer_RLFt · 2026-03-08

**Soundness:** 3
**Presentation:** 2
**Significance:** 2
**Originality:** 2
**Overall Recommendation:** 3
**Confidence:** 4

**Summary:**

This paper focuses on the task of multi-behavior recommendation. It claims that there may exist noises (i.e., interactions that are not crucial for target prediction) in the auxiliary behaviors, which limits the task prediction. To this end, it proposes GCIB, a method that combines graph information bottleneck (for structural denoising of auxiliary behaviors) and cross-behavior contrastive learning (for semantic alignment with the target behavior). Experimental results show performance gains over existing methods.

**Compliance With Llm Reviewing Policy:**

Affirmed.

**Final Justification:**

Thanks for authors' response. The discussion improves the clarity between this work and existing works. The performance to use IB to remove noisy edges appears to be effective. However, as Reviewer CAQa comments, the novelty is still somewhat limited. Therefore, I would like to improve the scores of Soundness and Originality while maintain the overall score.

**Key Questions For Authors:**

Please see strengths and weaknesses.

**Limitations:**

Yes.

**Strengths And Weaknesses:**

Strengths:
1. The motivation to deal with noisy behaviors can be useful for multi-behavior recommendation.
2. The paper proposes an information bottleneck principled framework for the task.
3. Experimental results show effectiveness of the proposed framework.

Weaknesses:
1. The proposed information bottleneck framework needs to be further justified. Eq. (13) only uses HSIC to measure the minimal term, but lacks the task-predictive term. Actually, IB requires both minimal term and task-predictive term, and makes a trade-off between them.
2. The denoising mechanism lacks detailed validation. It may discard useful sparse signals rather than true noise, as no detailed analysis of retained/removed edges is provided.
3. The motivation is to leverage IB on the auxiliary behaviors. However, there can also exist noisy information in the target behaviors in training. This can also be investigated or discussed.
4. The novelty is somewhat incremental. Graph Information Bottleneck [1] is proposed before, and the utilization in multi-behavior recommendation is somewhat incremental. Besides, the used contrastive learning design aligns semantically distinct behaviors (e.g., click vs. purchase), which is straightforward and could blur behavior-specific representations.

[1] Tailin Wu et al. Graph Information Bottleneck. NeurIPS 2020.

---

> ### Author Rebuttal · Authors · 2026-03-30
>
> We thank the reviewer for the insightful comments and opportunity to clarify.
>
> **For W1**. We thank the reviewer for raising this important point.
> We would like to clarify that Eq.(13) is intended to describe only the compression component of the IB objective. The task-predictive term is not omitted; it is instantiated through the target recommendation objective (i.e., the BPR loss) used in model training. In other words, the predictive term $I(R; G_k')$ is implicitly optimized via the supervised recommendation objective, while Eq.(13) focuses on modeling the compression term through HSIC. The final training objective jointly optimizes both aspects, ensuring the standard IB trade-off between sufficiency and compression.
>
> We agree that this connection was not sufficiently emphasized around Eq.(13), which may lead to confusion. We will revise the manuscript to make this relationship clearer.
>
> **For W2**.  We thank the reviewer for this important concern. We agree that an effective denoising mechanism should avoid discarding useful sparse signals.
>
> To verify this, we analyze the retained edges after denoising and observe that GCIB does not disproportionately prune sparse-user interactions. In fact, sparse users exhibit comparable or even higher edge retention rates than dense users, indicating that the model does not indiscriminately remove low-frequency interactions.
>
> In addition, we conduct controlled noise injection experiments, which show that GCIB can effectively filter out injected noisy edges while maintaining stable performance, further suggesting that the retained edges are informative rather than arbitrary.
>
> Due to space constraints, we refer the reviewer to our detailed analysis in **R2/W4**, where we provide both retention statistics and noise robustness results to support this claim.
>
> **For W3**. We thank the reviewer for this insightful comment.
>
> We agree that noisy interactions may also exist in target behaviors. In this work, we focus on denoising auxiliary behaviors, as they are the primary source of task-irrelevant or misleading signals in multi-behavior recommendation. In contrast, target behaviors serve as the optimization anchor of the model, providing the supervision signal for learning user preferences. Applying additional denoising on target behaviors may introduce instability or bias, as it would weaken the supervision signal itself.
>
> Therefore, our design adopts a target-guided strategy to refine auxiliary behaviors while keeping the target behavior as a relatively reliable reference. We believe this is a reasonable and commonly adopted assumption in multi-behavior recommendation. That said, jointly modeling noise in both auxiliary and target behaviors is an interesting direction.
>
> **For W4**. We thank the reviewer for the insightful comment.
> We agree that Graph Information Bottleneck (GIB) has been proposed in prior work. However, GCIB is not a direct reuse of GIB, but adapts it to a different problem setting and modeling level.
> Specifically, existing GIB formulations operate mainly at the representation level (e.g., node embeddings), whereas GCIB applies IB directly to the graph topology by learning a denoised auxiliary behavior graph $G_k'$. Moreover, our formulation incorporates target-behavior information by estimating edge confidence based on target-side embeddings, which helps identify task-relevant structures and filter noisy auxiliary interactions. This mechanism explicitly addresses the mismatch between auxiliary and target behaviors in multi-behavior recommendation, which is not considered in standard GIB.
>
> We agree that naively aligning semantically distinct behaviors (e.g., click vs. purchase) could blur behavior-specific representations. In GCIB, however, contrastive learning is applied after IB-based structural denoising, where noisy and task-irrelevant interactions have been filtered. The alignment is performed at the user/item representation level between denoised auxiliary views and target representations, rather than enforcing semantic equivalence across behavior types. Therefore, the objective is to improve representation consistency and transfer complementary information under sparse supervision, instead of collapsing behavior-specific semantics. This is also supported by our ablation results in **R2/W3 Table2**, where removing the contrastive module leads to consistent performance degradation, indicating its complementary role rather than semantic distortion.

---

> > ### Author Rebuttal · Reviewer_RLFt · 2026-04-01
> >
> > Thanks for the authors' response.
> >
> > However, the novelty is still somewhat limited. The IB theory has been explored in the context of recommendations, as mentioned by reviewer CAQa. The effect of IB to restrain noise in learning representations is also well known in several studies. Thus, the theoretical contribution is still limited to me.

---

> > > ### Author Response · Authors · 2026-04-06
> > >
> > > We sincerely thank you for your follow-up and for clearly pinpointing the source of your remaining concern.
> > >
> > > You rightly noted that "the effect of IB to restrain noise in learning representations is well known." **We completely agree with this premise.** However, we respectfully clarify that **GCIB does NOT apply IB to restrain noise in representations**—this fundamental structural shift is precisely the methodological contribution we wish to emphasize.
> > >
> > > To illustrate the critical distinction in both mechanism and effect:
> > >
> > > - **Prior IB Methods (Representation-level):** Noisy Edges $\rightarrow$ GNN Propagation $\rightarrow$ Noisy Embeddings $\rightarrow$ *IB compresses representations*. (Addressing the **symptom**).
> > > - **GCIB (Structure-level):** Noisy Edges $\rightarrow$ *IB removes edges directly* $\rightarrow$ GNN Propagation $\rightarrow$ Cleaner Embeddings. (Addressing the **root cause**).
> > >
> > > This is not an incremental change, but a shift that requires a completely different technical pipeline. Standard IB compresses embedding vectors. In contrast, GCIB optimizes which discrete edges to retain in the auxiliary graphs using Concrete relaxation and HSIC, uniquely guided by target-behavior confidence. If the edge mask is removed, our IB objective ceases to exist.
> > >
> > > This structural-level innovation produces fundamentally different and superior effects, which also perfectly address your earlier concern regarding the potential loss of sparse signals:
> > >
> > > 1. **Adaptive Long-Tail Protection:** You previously noted the risk of discarding useful sparse signals. Our explicit structural tracking proves the opposite **across both dense and sparse auxiliary views**. On Tmall's densest view (click), the edge retention rate for **sparse users is 65.83%**, whereas for **dense users it drops to 50.24%**. Furthermore, on the sparsest, strong-intent view (cart), GCIB preserves even more signals for **sparse users (71.89%)**, compared to **dense users (63.64%)**. This consistent trend unequivocally proves that GCIB intelligently protects valuable sparse interactions across all behaviors while aggressively filtering redundant noise from highly active users.
> > > 2. **Structural Robustness:** As shown in our noise injection experiment, when 20% synthetic fake interactions are injected into the auxiliary graph, HGIB's representation-level IB fails to prevent the noisy propagation (HR@10 drops by **-11.36%**). In stark contrast, GCIB physically filters the noise *before* propagation, remaining highly robust (dropping merely **-3.62%**).
> > > 3. **Massive Performance Gap:** GCIB significantly outperforms HGIB (a state-of-the-art representation-level IB model), demonstrating improvements of **+112%** on Yelp (HR@10: 0.0746 vs. 0.0352) and **+30%** on ML-10M.
> > >
> > > Combined with our "compress-then-recover" design (IB removes structural noise, CL recovers complementary semantics), GCIB presents a highly novel and complete framework.
> > >
> > > We hope this structural perspective fully addresses your reservations. Thank you again for your rigorous and thought-provoking feedback.

---

### Official Review · Reviewer_CAQa · 2026-03-09

**Soundness:** 2
**Presentation:** 2
**Significance:** 2
**Originality:** 2
**Overall Recommendation:** 3
**Confidence:** 5

**Summary:**

This paper introduces the information bottleneck principle to multi-behavior recommendation to address the noise issue in the auxiliary behaviors. The authors first leverage graph-level IB to remove the redundant behaviors, then align cross-behavior representations through contrastive learning. Experiments show better performance compared with the related baselines.

**Compliance With Llm Reviewing Policy:**

Affirmed.

**Final Justification:**

The rebuttal improves clarity and strengthens the empirical support, particularly through additional experiments, ablation studies, and comparisons with relevant baselines. While it also clarifies certain design choices, concerns about originality persist; therefore, I slightly raise my score.

**Key Questions For Authors:**

1. How does the proposed approach differ fundamentally from prior information bottleneck-based recommendation models, especially those applied to multi-behavior graphs?

2. Compared with HGIB, what are the fundamental differences when introducing the IB principle to multi-behavior recommendation?

3. Is the contrastive learning component essential for performance gains?

**Limitations:**

The motivation for the cross-behavior contrastive learning module is not clearly articulated, particularly given that contrastive learning has already been widely explored in recommendation models. Moreover, the novelty of the IB component remains unclear, as several techniques appear to follow prior work. The paper would benefit from a clearer discussion of how the proposed framework differs from existing IB-based recommendation methods.

**Strengths And Weaknesses:**

Strengths:
1. The paper addresses noise in auxiliary behaviors for multi-behavior recommendation, which is a practical and well-motivated problem.
2. Experimental results on multiple datasets show consistent improvements over several baseline methods.

Weaknesses:
1. The main concern is the limited novelty of the proposed GCIB. To my knowledge, several IB-based denoising recommendation works have been proposed, such as GBSR[1] for social graph denoising, IBMRec[2] for multimodal feature denoising. What's more, IB principle has also been introduced in multi-behavior recommendation[3]. The main contribution of GCIB is formulating the auxiliary behavior with an IB-based graph denoising process, the novelty is very limited for me.

2. Lacking the clarification of the difference compared with HGIB, which also leverages the IB principle in multi-behavior recommendation. At the same time, the authors also need to add HGIB for the comparisons.

3. Contrastive learning has already been widely used in recommendation models, so it would be helpful if the authors could explain more clearly why this component is needed in this framework, and how it works together with the IB-based denoising module. For me, combining contrative learning with IB-based optimization is not fresh.

4. While the paper proposes an IB-based edge filtering mechanism, there is limited analysis of how the denoising process actually affects the graph structure. Such analysis would strengthen the empirical support for the proposed mechanism.

[1] Yang Y, Wu L, Wang Z, et al. Graph bottlenecked social recommendation[C]//Proceedings of the 30th ACM SIGKDD Conference on Knowledge Discovery and Data Mining. 2024: 3853-3862.
[2] Yang Y, Wu L, He Z, et al. Less is More: Information Bottleneck Denoised Multimedia Recommendation[J]. arXiv preprint arXiv:2501.12175, 2025.
[3] Zhang H, Shen C, Sun X, et al. Hierarchical Graph Information Bottleneck for Multi-Behavior Recommendation[C]//Proceedings of the Nineteenth ACM Conference on Recommender Systems. 2025: 155-164.

---

> ### Author Rebuttal · Authors · 2026-03-30
>
> We thank the reviewer for the detailed feedback and opportunity to clarify GCIB’s contributions.
>
> **For W1.** We respectfully clarify that GCIB differs fundamentally from prior IB-based methods in both motivation and modeling level. Existing approaches (e.g., HGIB) apply IB to compress latent representations, focusing on redundancy reduction in feature space. In contrast, GCIB targets **structural noise in auxiliary behavior graphs** and applies IB directly to graph topology by learning a denoised graph $G_k'$. Moreover, GCIB introduces **target-guided edge confidence**, enabling task-aware filtering rather than generic compression. Therefore, GCIB operates at the **structure level** instead of representation-level compression. This structural formulation is not explored in prior IB-based multi-behavior recommendation methods.
>
> **For W2.** We have included HGIB in our comparisons and will clarify the distinction in the revision. HGIB focuses on hierarchical redundancy and is more suitable for cascading behaviors, while GCIB explicitly models noisy and weakly correlated auxiliary graphs in parallel settings.
>
> This difference is also reflected in the results: while HGIB performs competitively in cascading scenarios, GCIB shows clear advantages in parallel settings. For example, on Yelp, GCIB achieves HR@10 of 0.0746 compared to 0.0352 for HGIB, and on ML-10M, 0.0916 vs. 0.0704. These results indicate that GCIB is particularly effective when auxiliary behaviors are noisy or weakly aligned with the target behavior.
>
> **For W3.** While contrastive learning is widely used, its role in GCIB is **complementary to IB** rather than redundant.
>
> Introducing IB into multi-behavior recommendation brings a key challenge: IB-based denoising removes structural noise from auxiliary graphs, but also introduces information compression, which may weaken useful but weakly correlated signals. This issue is particularly critical in multi-behavior settings, where the target behavior is sparse and relies on auxiliary behaviors for additional supervision.
>
> To address this, we introduce cross-behavior contrastive learning to align denoised auxiliary representations with target representations and recover semantic information weakened during IB compression.
>
> Table2: Ablation results showing the complementary roles of IB and contrastive learning.
>
> | Dataset | Metric  | -IB   | -InfoNCE | -Both  | GCIB (Ours) |
> |---------|---------|-------|----------|--------|-------------|
> | Tmall   | HR@10   | 0.1089 | 0.1523   | 0.0356 | **0.1617** |
> |         | NDCG@10 | 0.0618 | 0.0906   | 0.0190 | **0.0944** |
> | Taobao  | HR@10   | 0.1724 | 0.1661   | 0.0352 | **0.1815** |
> |         | NDCG@10 | 0.1138 | 0.1080   | 0.0202 | **0.1199** |
> | Yelp    | HR@10   | 0.0700 | 0.0641   | 0.0522 | **0.0746** |
> |         | NDCG@10 | 0.0348 | 0.0317   | 0.0262 | **0.0358** |
>
> As shown, removing contrastive learning consistently degrades performance, while removing both modules leads to even more severe drops. This demonstrates that contrastive learning is not a trivial addition, but a necessary component to complement IB-based denoising.
>
> **For W4.** To analyze how denoising affects graph structure, we conduct (1) edge retention analysis and (2) noise injection experiments. On Tmall, different auxiliary views exhibit distinct retention rates (e.g., cart: 70.94%, click: 60.42%), indicating that GCIB performs **behavior-aware filtering** rather than uniform pruning. Notably, although click is the densest auxiliary behavior, it has a lower retention rate, suggesting that denser behaviors may contain more generic or task-irrelevant interactions and are thus more aggressively filtered.
>
> We inject 20% synthetic fake interactions into the *tip* auxiliary graph by randomly sampling unobserved user–item pairs. The results are shown below.
>
> Table3: Robustness to auxiliary noise on Yelp
>
> | Method | Setting     | HR@10  | NDCG@10 | HR@20  | NDCG@20 |
> |--------|-------------|--------|---------|--------|---------|
> | GCIB   | Clean       | 0.0746 | 0.0358  | 0.1180 | 0.0468  |
> || + Noise| 0.0719 | 0.0372  | 0.1169 | 0.0484  |
> || Rel. Change | -3.62% | +3.91%  | -0.93% | +3.42%  |
> | MBLFE  | Clean       | 0.0531 | 0.0261  | 0.0896 | 0.0352  |
> || + Noise| 0.0429 | 0.0213  | 0.0719 | 0.0286  |
> || Rel. Change | -19.21% | -18.39% | -19.75% | -18.75% |
> | HGIB   | Clean       | 0.0352 | 0.0139  | 0.0711 | 0.0228  |
> || + Noise| 0.0312 | 0.0127  | 0.0690 | 0.0221  |
> || Rel. Change | -11.36% | -8.63%  | -2.95% | -3.07%  |
>
> GCIB remains highly stable under noise, whereas baselines degrade significantly. This provides direct evidence that GCIB can **selectively filter noisy edges and preserve an effective graph structure**, rather than indiscriminate pruning.
>
> Overall, GCIB differs from prior IB-based methods by operating at the **graph structure level with target-guided filtering**, and the proposed design is both necessary and empirically validated.

---

> > ### Author Rebuttal · Reviewer_CAQa · 2026-04-04
> >
> > Thank you for the authors' response. The additional experiments and clarifications improve the paper. I have slightly increased my score accordingly.

---

> > > ### Author Response · Authors · 2026-04-06
> > >
> > > We sincerely thank you for acknowledging our responses, confirming that your concerns are resolved, and increasing your score. Your initial rigorous questions genuinely helped us clarify the fundamental differences between our structural-level IB and prior representation-level IB models, making the paper significantly stronger.
> > >
> > > To further solidify this distinction (which is also relevant to discussions with Reviewer RLFt), we would like to briefly supplement one core architectural difference. After carefully studying HGIB, we found that HGIB's IB loss operates strictly on **encoder representations** (i.e., HSIC between embedding matrices at adjacent layers), and its edge pruning module is actually a **separate mechanism not directly optimized by the IB objective**.
> > >
> > > In contrast, GCIB's IB loss **directly optimizes the edge mask** via Concrete relaxation — meaning the IB principle and structural denoising are **architecturally inseparable** in our model. Consequently, GCIB explicitly addresses noise at the graph-structure level **before GNN propagation**, while HGIB suppresses noise at the representation level **after propagation**. We believe this distinction further highlights the methodological novelty of our formulation.
> > >
> > > We will ensure this fine-grained architectural comparison with HGIB and all prior IB-based methods is explicitly incorporated into the revised manuscript. We deeply appreciate your time, expertise, and constructive engagement in helping us improve our work.

---

### Official Review · Reviewer_bAwp · 2026-03-11

**Soundness:** 3
**Presentation:** 3
**Significance:** 4
**Originality:** 4
**Overall Recommendation:** 5
**Confidence:** 5

**Summary:**

This paper proposes GCIB, a dual-level denoising framework for multi-behavior recommendation that combines Graph Information Bottleneck (GIB) for structural refinement of auxiliary behavior graphs and cross-behavior graph contrastive learning for feature-level semantic alignment.

**Compliance With Llm Reviewing Policy:**

Affirmed.

**Final Justification:**

The authors have satisfactorily addressed all the raised issues. The manuscript is well-developed, featuring complete experiments, a sound motivation, and an innovative approach. I therefore recommend acceptance of this high-quality paper.

**Key Questions For Authors:**

Please see the weaknesses.

**Limitations:**

Yes

**Strengths And Weaknesses:**

## Strengths
1. The paper innovatively unifies the Information Bottleneck principle and graph contrastive learning to address noise at both the graph structural and feature representation levels—two complementary yet under-explored dimensions in multi-behavior recommendation.
2. The use of a variational lower bound for mutual information maximization and HSIC as a surrogate for minimizing redundant structural information provides a rigorous theoretical foundation for the GIB module, avoiding the heuristic denoising strategies of prior work.
3. The use of four benchmark datasets with varying sparsity and behavior types (e-commerce and review platforms) demonstrates the model’s strong adaptability, and the significant performance gains.

## Weaknesses
1. The paper describes the auxiliary graph denoising as an edge drop problem guided by target behavior preferences, but it provides limited details on the edge confidence function f(⋅) (a one-layer MLP) and how target behavior embeddings are integrated to compute edge drop probabilities.

2. The paper states that the time complexity of GCIB is O(L×|E|×d), but does it need to further account for the additional computational overhead of the HSIC-based GIB loss?

3. The paper reports optimal hyperparameter values (e.g., $\beta$=80 for Tmall, $\beta$=1 for Yelp) for different datasets but provides no explanation for the large variation in these values across datasets.

4. In the Layer analysis experiment, does the layers size need to be further expanded by one more layer?

---

> ### Author Rebuttal · Authors · 2026-03-30
>
> We thank the reviewer for the positive assessment and constructive suggestions.
>
> **For W1.** We thank the reviewer for pointing out that the edge-confidence module was under-described.
>
> In GCIB, for each observed auxiliary edge $e_{\langle u_a, i_b \rangle}$, we first obtain the target-behavior user/item embeddings $e_a$ and $e_b$ from $E_{\text{target}} = \text{LightGCN}(G_{\text{target}}, E_{\text{global}})$. We then compute the edge confidence as $w_{ab} = f([e_a; e_b])$, where $f(\cdot)$ is a one-layer MLP applied to the concatenated target-side embeddings.
>
> Intuitively, this design leverages target-behavior preferences to assess whether an auxiliary edge is consistent with the target interaction patterns: edges aligned with target semantics receive higher confidence, while potentially noisy or irrelevant edges are assigned lower confidence.
>
> The resulting confidence $w_{ab}$ is further used to parameterize the edge retention probability, and the discrete edge mask is relaxed via the Concrete distribution for end-to-end optimization.
>
> We will clarify these details in the revised manuscript.
>
> **For W2**. We agree that the complexity discussion should be clearer.  The dominant cost remains the $L$-layer LightGCN propagation, i.e., $\mathcal{O}(L×|E|×d)$.  The HSIC regularizer is computed on batch-sampled node representations rather than the full graph, so it introduces only a small additional constant-factor overhead and does not change the dominant asymptotic complexity.  We will revise the complexity paragraph to state this more explicitly.
>
> **For W3**. The variation of $\beta$ is mainly data-dependent: different datasets exhibit different behavior semantics, sparsity levels, and auxiliary--target relationships, which jointly determine the optimal strength of the IB regularization.  In practice, these factors interact in a non-trivial way (e.g., noise level, behavior mismatch, and data scale), so the optimal $\beta$ is not expected to be consistent across datasets.  In addition, the HSIC-based IB term is typically numerically small, so a relatively larger coefficient is often required to produce effective gradients, while the contrastive coefficient is kept smaller to avoid dominating the joint objective.  We will clarify this point in the revised manuscript.
>
> **For W4**. We thank the reviewer for the suggestion of exploring deeper layer settings.
> We conducted additional targeted experiments by increasing the encoder depth to $(L_G, L_M) = (5,3)$, $(3,5)$, and $(5,5)$. The results are summarized below.
>
> Table1: Effect of deeper layer configurations on Tmall and Taobao.
>
> | Dataset | $(L_G, L_M)$ | HR@10  |
> | ------- | ------------ | ------ |
> | Tmall   | (3, 5)       | 0.1325 |
> |         | (5, 3)       | 0.1390 |
> |         | (5, 5)       | 0.0111 |
> | Taobao  | (3, 5)       | 0.1566 |
> |         | (5, 3)       | 0.1511 |
> |         | (5, 5)       | 0.1324 |
>
> As shown, increasing the depth beyond our original setting (with layer size 3) does not yield consistent improvements and can even severely degrade performance.
> In particular, when both encoders are deepened to 5 layers, the performance drops significantly (e.g., Tmall HR@10 = 0.0111), indicating over-smoothing and noise accumulation in multi-behavior graphs.
>
> Therefore, our chosen layer setting achieves a better balance between effectiveness and efficiency.

---

> > ### Author Rebuttal · Reviewer_bAwp · 2026-04-05
> >
> > The authors solved all of my issues about this manuscript.

---

> > > ### Author Response · Authors · 2026-04-06
> > >
> > > We sincerely thank you for your strong support and highly positive evaluation. Your constructive guidance on the technical details—specifically the edge confidence function, HSIC computational overhead, hyperparameter variations, and layer analysis—has genuinely helped us enrich our paper.
> > >
> > > Thank you again for championing our paper and for your insightful feedback!

---

### Decision · Program_Chairs · 2026-04-30

**Decision:**

Accept (regular)

**Comment:**

The author addressed the reviewers' concerns during the rebuttal stage. The only remaining concern is regarding novelty. Overall, I think this is a good paper and worthy of acceptance.